chemical physics/computational chemistry

triallyl isocyanurate, UV radiation cross-linking polyethylene process, transition states

**Authors for correspondence:**
Hui Zhang
e-mail: hust_zhanghui11@hotmail.com,
huizhang@hrbust.edu.cn
Hong Zhao
e-mail: hongzhao@hrbust.edu.cn
Baozhong Han
e-mail: hbzhlj@163.com

This article has been edited by the Royal Society of Chemistry, including the commissioning, peer review process and editorial aspects up to the point of acceptance.

# Further discussion on the reaction behaviour of triallyl isocyanurate in the UV radiation cross-linking process of polyethylene: a theoretical study

Hui Zhang[1], Yan Shang[1], Hong Zhao[1], Xuan Wang[1], Baozhong Han[1,2] and Zesheng Li[3]

[1]Key Laboratory of Engineering Dielectrics and Its Application of Ministry of Education and College of Chemical and Environmental Engineering, Harbin University of Science and Technology, Harbin 150080, People's Republic of China
[2]Shanghai Qifan Cable Co., Ltd, Shanghai 200008, People's Republic of China
[3]Key Laboratory of Cluster Science of Ministry of Education and School of Chemistry, Beijing Institute of Technology, Beijing 100081, People's Republic of China

HZ, 0000-0002-3833-0764

Further theoretical investigation on the reaction behaviour of triallyl isocyanurate (TAIC) in the UV radiation cross-linking process of polyethylene (PE) is accomplished by density functional theory for high voltage cable insulation materials. The reaction potential energy information of the 13 reaction channels at B3LYP/6–311 + G($d$,$p$) level are identified. These have been explored that the TAIC take part in the reaction behaviour on ground state during UV radiation cross-linking process and TAIC intra-molecular isomerization reaction itself. In addition, the results show that the effect of multiplication and acceleration for the cross-linking reaction of trimethylopropane trimethacrylate (TMPTMA) would be better than that of TAIC. It has further clarified the reasons why UV radiation cross-linking reaction of PE had been initiated by benzophenone (Bp), and the TAIC or TMPTMA needed to take part. The results obtained in the present study could directly guide both the optimization of UV radiation cross-linking PE process and the development of the insulation material of high-voltage cable in real application.

# 1. Introduction

Cross-linked polyethylene (XLPE) has been widely applied as high voltage cable insulation materials due to its excellent electrical and mechanical properties. To produce polyethylene (PE) insulation materials, Ultraviolet (UV) radiation cross-linking technology was first proposed by Oster in the 1950s and further developed to be applicable to industrial XLPE production by Rånby and co-workers [1–4]. Compared with the traditional peroxide process, it has many advantages. UV energy can easily go through the insulation wall and induce the cross-linking with the help of photo-initiator. Experimental results showed that the rate for cross-linking of PE with photo-initiator benzophenone (Bp) by UV radiation is lower than that with multi-functional cross-linker 1,3,5-triallyl isocyanurate (TAIC) participating. We recently [5] investigated the role of TAIC in UV radiation cross-linking process of PE using density functional theory (DFT) calculations [6]. The reaction potential barriers for the formation of Pe2 radical by Bp and TAIC are 0.20 and 1.70 eV, respectively [5,7]. The generated PTAIC radical could occur quickly intra-molecular isomerization reaction with the calculated potential barrier of 0.66 eV [5] to yield PTAIC1 radical, which has three sites for cross-linking with PE radical. The molecular formula of the mentioned abbreviations of radical (Pe2, PTAIC and PTAIC1) are listed in table 1. These results explain the experiential phenomenon successfully; this is why the UV radiation cross-linking reaction of PE can be initiated by Bp, and the cross-linking rate grade has been multiplied by TAIC. The rate of the PE UV radiation cross-linking can reach minute grade only with the photo-initiator Bp, while up to second grade with TAIC participating [5]. In addition, the different cross-linker shows different reactivity, for example, trimethylopropane trimethacrylate (TMPTMA) could increase both the reaction rate and cross-linking degree [8]. Rånby and co-workers reported that two cross-linkers, triallyl cyanurate (TAC) and TAIC, have almost identical effects on the cross-linking process [4]. Structure transformation was involved in the polymerization of these multi-functional monomers [9]. The differential thermal analysis (DTA) curves and infrared spectral data indicate that TAC can be transformed to TAIC by the rearrangement of the three allyl groups. By now, no theoretical work has been addressed on these issues. In this work, we selected 4-methylheptane (Pe) as model molecule of cross-linkable PE to investigate the further reaction behaviour of the ground state TAIC, intra-molecular isomerization reaction of TAIC to form TAC, the comparison of multiplication effect of TMPTMA for the cross-linking reaction with TAIC using DFT calculations [6]. The transition states and the products are abbreviated to the corresponding TS and P, respectively, and reaction channels are represented by R. The molecular formula and corresponding abbreviations of the studied molecules are listed in table 1. The obtained results are important not only for the optimization of UV radiation cross-linking PE process or the selection of multi-functional cross-linker, but also for the development of high voltage cable in real application.

# 2. Computational methods

The geometry optimization and frequency calculation of the studied molecules on the ground state were performed by using the B3LYP [10–13] functional with the $6–311+G(d,p)$ basis set, and this level was confirmed to obtain very reliable results for the current study [14]. On the basis of the optimized geometries, the energy gap ($E_g$) between the highest occupied molecular orbital (HOMO) and the lowest unoccupied molecular orbital (LUMO) was calculated. The ionization potential (IPs) and the electron affinity (EAs) can be defined as that in the previous study [14]. $v$ and $a$ represent vertical energy and adiabatic energy, respectively. The time-dependent density functional theory (TDDFT) [15,16] was employed to calculate the three lowest excitation energies ($S_1$, $S_2$ and $S_3$) of Bp, TAIC and TMPTMA molecules on the basis of the optimized ground state geometries. Intrinsic reaction coordinate (IRC) calculation with a gradient step-size of 0.05 (amu)$^{1/2}$ bohr was performed to obtain the minimum energy path (MEP). All the calculations in the present study were performed using GAUSSIAN09 program package [17].

# 3. Results and discussion

## 3.1. Stationary point geometries

The 13 possible chemical reaction equations of TAIC and TMPTMA at the ground state with Pe or Bp are listed in table 2. The optimized structures of the reactants, products and transition states involved in these reaction channels at the B3LYP/6–311+G(d,p) level are presented in figure 1. The optimized standard

**Table 1.** Molecular formula and corresponding abbreviation of mentioned molecules.

| molecular formula | ab. | molecular formula | ab. |
|---|---|---|---|
|  | Pe |  | Pe2 |
|  | Bp |  | Pe4 |
|  | TAIC |  | PBp |
|  | TMPTMA |  | PTAIC |
|  | PTMPTMA |  | PTAIC1 |
|  | TAC |  | PTMPTMA1 |
|  | TAC1 |  | PTMPTMA2 |
|  | TAC2 |  | PTMPTMA3 |

orientation of equilibrium geometries of the transition states in the electronic supplementary material (ESM-1). The key bond lengths and the imaginary frequency values in the stationary points are also listed in table 2 as well as the corresponding number of reactions. The transition states are all confirmed by normal-mode analysis to have only one imaginary frequency, which corresponds to the stretching modes of coupling between breaking and forming bonds. In table 2, one can see that the two transition state structures TSPBpTAIC and TSPBpTMPTMA have a common character that the elongation of the breaking O–H bonds in PBp molecule are larger than that of the corresponding forming O–H bonds in equilibrium PTAIC and PTMPTMA molecules, respectively, indicating that the two hydrogen abstraction reactions are all product-like, i.e. those reaction channels will proceed via 'late' transition states, which is consistent with Hammond's postulate [18], applied to an endothermic reaction.

**Table 2.** Optimized bond lengths of breaking/forming bonds for transition state, reactants and products (in angstrom), together with the calculated breaking/forming bond frequencies (in $cm^{-1}$), the reaction Gibbs free energies at 298 K ($\Delta G$) and the Gibbs potential barrier heights ($\Delta G^{\neq}$) with zero-point energy (ZPE) corrections and the dissociation energies of breaking bond in reactants (in eV) at the B3LYP/6−311 + G($d,p$) level.

| number | reaction equation | reactant | b/f | product | freq. | $\Delta G^{\neq}$ | $\Delta G$ | $D_{298}$ |
|---|---|---|---|---|---|---|---|---|
| ① | | 1.100 | 1.228/1.395 | 0.964 | 835 $i$ | 0.60 | −0.48 | 3.91 |
| ② | | 1.100 | 1.187/1.492 | 0.950 | 254 $i$ | 2.18 | 0.99 | 3.33 |
| ③ | | 0.964 | 1.666/– | — | 916 $i$ | 1.48 | 1.04 | 1.60 |
| ④ | | 0.964 | 1.537/1.023 | 0.968 | 361 $i$ | 1.55 | 1.32 | 1.60 |
| ⑤ | | 0.964 | 1.278/1.128 | 0.965 | 1455 $i$ | 1.15 | 0.19 | 1.60 |
| ⑥ | | — | –/1.508 | 0.965 | 1198 $i$ | 0.16 | −0.85 | — |
| ⑦ | | 0.965 | 1.250/1.290 | 1.125 | 1879 $i$ | 1.67 | 0.91 | 1.59 |
| ⑧ | | 1.125 | 1.332/1.583 | 1.094 | 1142 $i$ | 0.31 | −1.42 | 0.68 |

(Continued.)

**Table 2.** (*Continued.*)

| number | reaction equation | reactant | b/f | product | freq. | $\Delta G^{\neq}$ | $\Delta G$ | $D^{\circ}_{298}$ |
|---|---|---|---|---|---|---|---|---|
| ⑨ |  | 0.965 | 1.323/1.499 | 1.094 | 2225 $i$ | 2.07 | — | 1.59 |
| ⑩ |  | 1.094 | 1.257/1.357 | 1.097 | 1713 $i$ | 1.16 | −0.63 | 2.01 |
| ⑪ |  | — | −/1.382 | 0.968 | 1310 $i$ | 0.73 | 0.27 | — |
| ⑫ |  | 0.968 | 1.148/1.497 | 1.093 | 2103 $i$ | 0.53 | −1.33 | 0.25 |
| ⑬ |  | 1.484 | 2.275/1.979 | 1.330 | 364 $i$ | 2.07 | 0.87 | 3.33 |
| ⑭ |  | 1.483 | 2.222/1.999 | 1.331 | 372 $i$ | 1.89 | 0.66 | 3.05 |
| ⑮ |  | 1.481 | 2.118/1.988 | 1.335 | 414 $i$ | 1.74 | 0.32 | 3.11 |

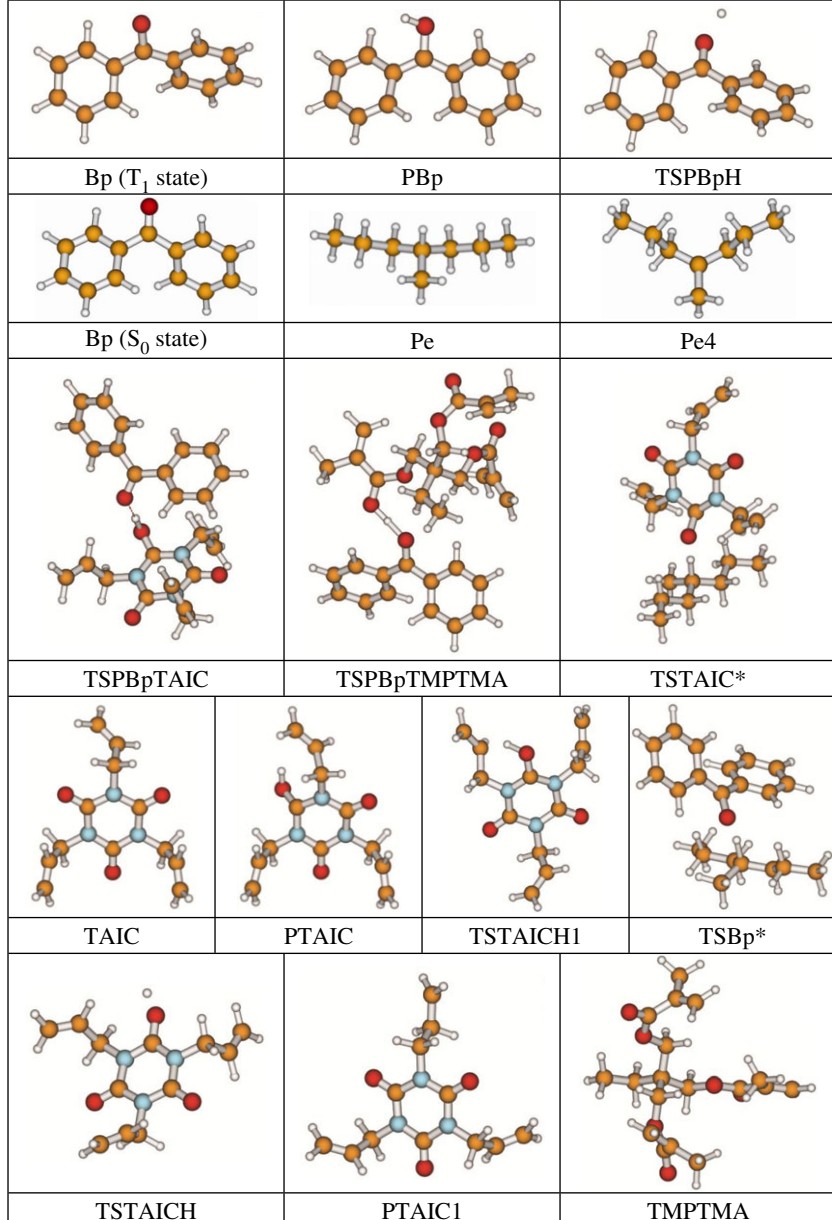

**Figure 1.** Optimized geometric structures of the studied molecules at the B3LYP/6–311 + G(d,p) level.

## 3.2. Energies: excitation energies, frontier MOs, IPs and EAs

The calculated excitation energies ($S_1$, $S_2$ and $S_3$) of Bp, TAIC and TMPTMA molecules at the B3LYP/ 6–311 + G(d,p) level are listed in table 3. The HOMO–LUMO energy gap ($E_g$), the ionization potential (IPs) and the electron affinity (EAs) of Pe, Bp, TAIC and TMPTMA molecules at the same level are also listed in table 3. The results show that TMPTMA has the lower electron excitation energy level than TAIC, so TMPTMA would be easier to be grafted to PE molecule by UV radiation than TAIC. TMPTMA, which has three ethylene groups, would act as the cross-linking agent and each ethylene group can connect one Pe molecule as the cross-linking site. IP and EA of the molecule are the important properties to estimate the ability of its reduction and oxidation, respectively. As shown in table 3, the calculated values of IP(a) and EA(a) for Bp are in good agreement with the corresponding experimental data [19] (in brackets), showing that the current computational level is reliable. For other molecules, the variation trends of the IP and EA are similar to those of the negative value of the corresponding HOMO and LUMO energies, respectively. The HOMO–LUMO energy gap ($E_g$) in Pe depends on the energy difference between molecule orbital $\sigma$ and $\sigma^*$ ($\sigma \rightarrow \sigma^*$) in carbon chain. The introduction of phenyl or heteroatom groups into the molecule, such as Bp, TAIC and TMPTMA, is propitious to electron

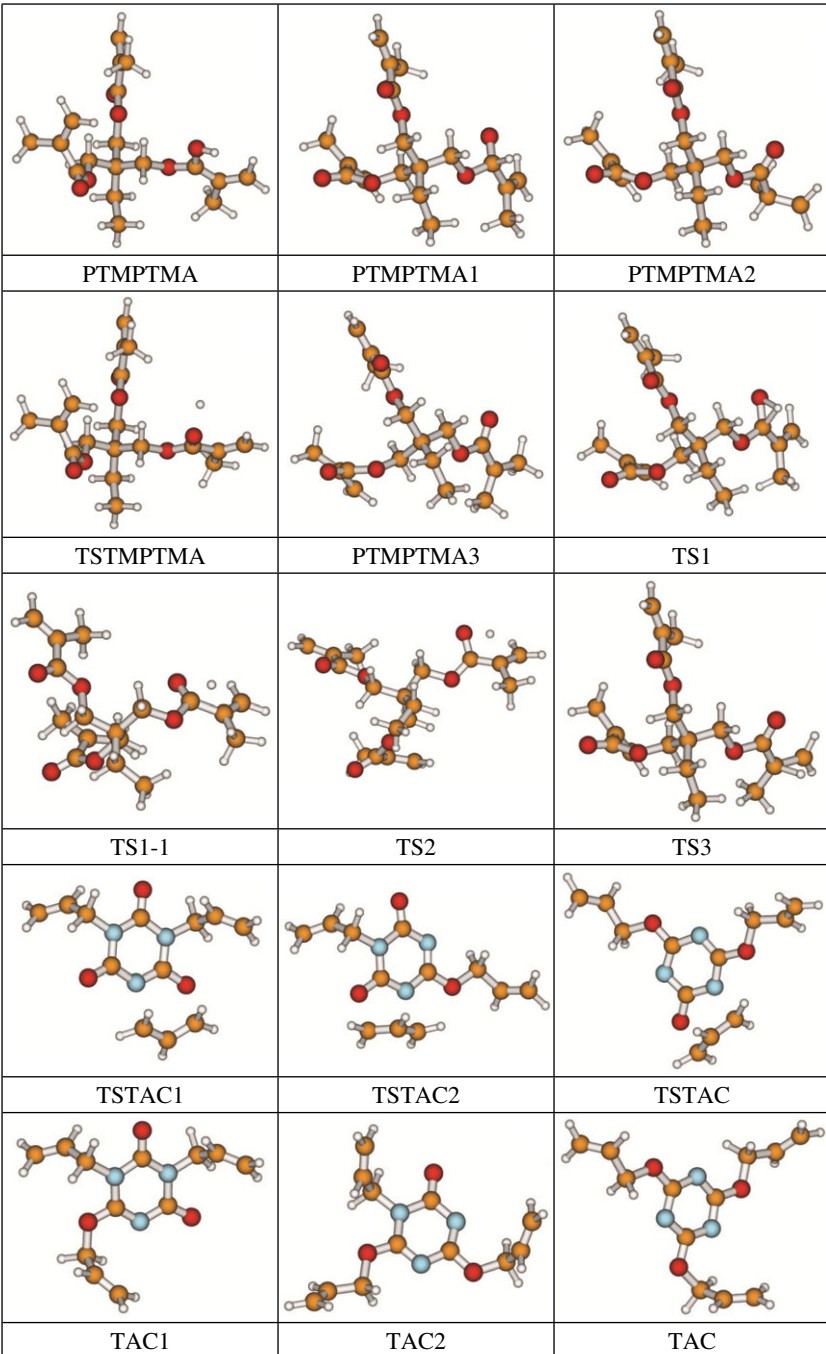

**Figure 1.** (*Continued.*)

dissociation because of the small ionization potentials in terms of Koopmans' theorem, the value of $E_g$ means the energy gap of $\pi \to \pi^*$. Thus, the $E_g$ values of Bp (4.90 eV), TAIC (6.90 eV) and TMPTMA (5.81 eV) are all smaller than that of Pe (8.38 eV). The $\pi$-$\pi$-$\pi$, $\pi$-$p$-$\pi$ and $\pi$-$\pi$-$p$ conjugative effects are formed in Bp, TAIC and TMPTMA respectively, as the result, it can also be seen that $E_g$ (Bp, TAIC or TMPTMA) < $E_g$ (Pe). The energy of electron transition becomes smaller with the decreasing of $E_g$. The conjugated molecules Bp, TAIC and TMPTMA with electron delocalization possess stronger capability of trapping electron than that of aliphatic chain, such as EA($a$) (Pe, −1.09) < EA($a$) (Bp, 0.73). The electric charge density on TAIC in $p$ orbit is larger than that of Bp, which results in TAIC having the weaker ability of accepting electron than that of Bp, EA($a$) (TAIC, −0.26) < EA($a$) (Bp). The number of oxygen atoms in TMPTMA is the most of all the compounds in table 3, and the electronegativity of oxygen atoms is higher than that of nitrogen and carbon atoms, so EA($a$) (Pe) < EA($a$) (TAIC) < EA($a$) (Bp) < EA($a$) (TMPTMA). The energy of electronic transition will be smaller when $E_g$ decreases, as mentioned above,

**Table 3.** Calculated excitation energies at the excited singlet state ($S_1$, $S_2$ and $S_3$), $E_g$, IPs and EAs ($v$ and $a$ represent vertical energy and adiabatic energy, respectively) at the group state of the studied molecules as well as the corresponding experimental data in brackets (in eV) at the B3LYP/6−311 + G(d,p) level.

| molecular formula | $S_n$ | excitation energy | $E_g$ | IP($a$) | IP($v$) | EA($a$) | EA($v$) |
|---|---|---|---|---|---|---|---|
| | 1 | 3.5991 | 4.90 | 8.64 | 8.67 | 0.73 | 0.50 |
| | 2 | 4.5270 | | (9.05) | | (0.69 ± 0.05) | |
| | 3 | 4.6060 | | | | | |
| | 1 | 5.7812 | 6.90 | 8.55 | 9.05 | −0.26 | −0.55 |
| | 2 | 5.7946 | | | | | |
| | 3 | 6.0665 | | | | | |
| | 1 | 4.8120 | 5.81 | 8.72 | 8.94 | 1.12 | 0.30 |
| | 2 | 4.9285 | | | | | |
| | 3 | 5.1907 | | | | | |
| | | | 8.38 | 9.41 | 10.03 | −1.09 | −1.10 |

so IP values of Bp, TAIC and TMPTMA are lower than Pe. These molecules can give rise to the collision ionization before PE chain is ionized, which can prevent the polymer matrix's degradation when they are doped in XLPE insulation composite product. They can also trap the 'hot electron', decrease the kinetic energy and dissipate energy, so the 'hot electrons' have not enough energy to break the C–C bonds of XLPE. As a result, they can inhibit the initiation and propagation of electrical tree in XLPE effectively and strengthen the electrical breakdown strength that XLPE can endure simultaneously. The XLPE insulation material would exhibit an elevated AC breakdown strength.

## 3.3. Energetics

The reaction Gibbs free energies at 298 K ($\Delta G$) and the Gibbs potential barrier heights ($\Delta G^{\neq}$) including zero-point energy (ZPE) corrections on $S_0$ state of the 13 reaction channels at the B3LYP/6–311 + G(d,p) level are also listed in table 2, as well as the relative breaking bond dissociation energies ($D_{298}^{g}$). The two reactions ① and ② at the excited triplet state $T_1^*$ are listed as reference [5,7]. The schematic diagrams of reaction progress of the 13 channels are plotted in the electronic supplementary material (ESM-2) as well as the corresponding number of reactions. The relevant schematic potential energy surfaces for the 13 reaction channels are plotted in figure 2. In figure 2, the other reactants have been introduced with plus sign '+'. The reaction channels without plus sign are dissociations or intra-molecular isomerizations. Bp would be excited from $S_0$ into the singlet excited state $S_1$ ($n$, $\pi^*$) and then transform into its excited triplet excited state $T_1$ ($n$, $\pi^*$) through intersystem crossing in the UV radiation cross-linking PE process, the Bp at $T_1$ state would initiate hydrogen abstraction reaction with Pe4 and form PBp and Pe4 radicals. The initiation behaviour of Bp was explored in detail in our previous work [7]. TAIC at $T_1$ state would react with Pe and form PTAIC and Pe4 radicals, then PTAIC would transport electron and proton and form PTAIC1 quickly; the double bond will be broken in PTAIC through intra-molecular isomerization reaction. The reaction potential barriers $\Delta E^{TS}$ with ZPE corrections at $T_1$ state for forming Pe4 radical by Bp via TSBp* and TAIC via TSTAIC* are 0.17 and 1.68 eV, respectively [5,7]; the Gibbs potential barrier heights ($\Delta G^{\neq}$) with ZPE corrections are 0.60 and 2.18 eV, respectively.

In this paper, further theoretical studies on the reaction behaviour of TAIC and the reaction of TMPTMA participating in have been accomplished. The reaction potential energy surface information on $S_0$ state of the 13 reaction channels has been investigated in detail at atomic and molecular levels. On $S_0$ state, the calculated result shows that the reaction Gibbs potential barrier of PBp with TMPTMA via TSPBpTMPTMA (1.15 eV) is lower than that of PBp with TAIC via TSPBpTAIC (1.55 eV) and hydrogen dissociation reaction channel via TSPBpH (1.48 eV), showing that the Gibbs potential barrier of the reaction channel RPBpTMPTMA is lower than that of the reaction channel RPBpTAIC. It is because that TMPTMA has the much stronger $\pi$-$\pi$-$p$ conjugation effect, much larger

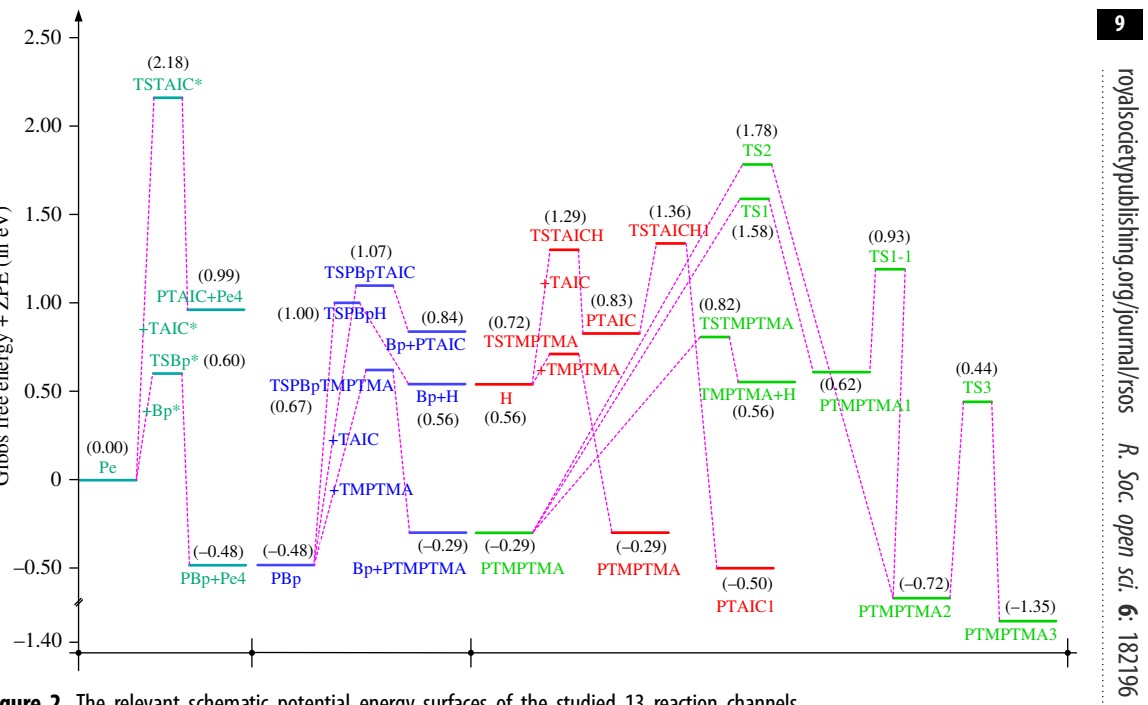

**Figure 2.** The relevant schematic potential energy surfaces of the studied 13 reaction channels.

electron delocalization, much lower electric charge density, much more O atom and much stronger electronegativity than TAIC. Thus, we infer that reaction rate of RPBpTMPTMA channel would be faster than that of the latter RPBpTAIC reaction channel.

The reaction channels of PTMPTMA to form PTMPTMA2 have two pathways, one is one-step hydrogen transfer reaction and the other is two-step hydrogen transfer reaction. The latter reaction has 0.40 eV lower Gibbs potential barrier than the former. The reaction potential barrier of PTAIC forming PTAIC1 via TSTAICH1 is only 0.66 eV [5]; the Gibbs potential barrier is 0.53 eV. Then, the PTAIC1 radical would be cross-linked with Pe radical at three reaction sites and get the stationary structures, the multiplication effect of cross-linking will be presented. The experimental phenomenon has been successfully explained that the rate can obtain minute grade when UV radiation cross-linking reaction of PE was initiated only by initiator Bp, while up to second grade when the multi-functional cross-linker TAIC participated in the cross-linking process [5]. The reaction Gibbs potential barrier of PTMPTMA forming PTMPTMA1 is 1.67 eV, and then the reaction Gibbs potential barrier of PTMPTMA1 forming PTMPTMA2 is 0.31 eV. The reaction Gibbs potential barrier of PTAIC forming PTAIC1 (0.53 eV) is lower than that of the PTMPTMA forming PTMPTMA2, because the breaking bond dissociation energy in PTAIC 0.25 eV is smaller than that of PTMPTMA 1.59 eV. Therefore, the H in PTAIC radical can dissociate easily and lead to more facile PTAICH1 radical with the lower energy barriers. The PTMPTMA2 radical would be cross-linked with Pe radical at three reaction sites to produce inactive products, in this way, this is in line with the experimental result as mentioned above, adding the multi-functional cross-linker TMPTMA would increase cross-linking degree [8].

It has been reported that the two cross-linkers TAC and TAIC have almost identical effects on the cross-linking process [4]. The two structures could accomplish transformation in which the TAC can transform to TAIC followed by rearrangement of the three allyl groups [9]. In this work, TAIC transformed to TAC through intra-molecular isomerization reaction itself has been also investigated at atomic and molecular levels. As shown in table 1 and electronic supplementary material, the reaction Gibbs potential barrier of TAIC forming TAC1 via TSTAC1 is 2.07 eV, TAC1 forming TAC2 via TSTAC2 is 1.89 eV and TAC2 forming TAC via TSTAC is 1.74 eV. With the increasing of number of the intra-molecular isomerized allyl groups, the reaction Gibbs potential barrier decreases generally.

Further work to account for reaction potential energy surface information of the negative molecular ions when electron injection, together with the following electronic transfer, as well as electronic mobility and interfaces energy etc. of multi-functional cross-linker in XLPE is underway. New experimental and theoretical efforts are required to select candidates of multi-functional cross-linker.

# 4. Conclusion

Further theoretical study on the TAIC reaction behaviour on ground state in the PE UV radiation cross-linking process has been carried out at the atomic and molecular levels. The reason why Bp initiates radical reaction and TAIC or TMPTMA accelerate cross-linking reaction has been further explained. The reaction channel RPBpTMPTMA forming PTMPTMA is more kinetically favourable than the reaction channel RPBpTAIC forming PTAIC based on the correlating reaction potential barrier analysis. The calculation results explain successfully the experiential phenomenon that the rate of the PE UV radiation cross-linking by TMPTMA is faster than that of TAIC. Further clarity of the reaction potential energy information on ground state is favourable for selecting and designing the perfect cross-linker, optimizing the PE UV radiation cross-linking process and developing the insulation material of high-voltage cable in real applications.

Data accessibility. This article does not contain any additional data.
Authors' contributions. H. Zhang carried out the geometry optimizations, participated in data analysis and drafted the manuscript; Y.S. participated in data analysis; H. Zhao and B.H. designed the study; X.W. carried out the statistical analyses; Z.L. conceived the study and participated in data analysis. All authors gave final approval for publication.
Competing interests. We declare we have no competing interests.
Funding. This research was funded by the National Natural Science Foundation of China (grant no. 51337002).
Acknowledgements. We thank Prof. Tierui Zhang (Key Laboratory of Photochemical Conversion and Optoelectronic Materials, Technical Institute of Physics and Chemistry (TIPC), Chinese Academy of Sciences (CAS), Beijing 100190, People's Republic of China) for his fruitful discussions and checking English.

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
