## [Reviewer comments · Royal Society Open Science]

Review History

RSOS-182196.R0 (Original submission)

Review form: Reviewer 1

Is the manuscript scientifically sound in its present form?

No

Are the interpretations and conclusions justified by the results?

Yes

Is the language acceptable?

No

Is it clear how to access all supporting data?

Yes

Do you have any ethical concerns with this paper?

No

Have you any concerns about statistical analyses in this paper?

No

Recommendation?

Major revision is needed (please make suggestions in comments)

Comments to the Author(s)

The authors present impressive amount of data leading eventually to better understanding of more effective polyethylene (PE) crosslinking mechanism. The basic species is 4-methylheptane (Pe in the author's notation) serving as a model of cross-linkable polyethylene. The data include the reaction energies, activation barriers, ionization and excitation energies and electron affinities of species supporting the effectivity of the crosslinking of polyethylene. From this point of view the paper is useful and interesting. Selected methods are adequate – for the geometry optimization, stationary point and transition states energies the common DFT/B3LYP method used in the present study is partly supported by previous successful applications to similar problems. Excitation energies are calculated by the TD DFT method. From this point of view no problem. The problem is that the presentation is rather poor. The paper is difficult to follow. Conclusions are not clearly supported by results. I would support publishing the paper only upon substantial improvement of the presentation.

1. I am lost in a jungle of abbreviations already while reading Part I., Introduction. Some are explained in Figure 1, mentioned much later (page 5), some remain unexplained at all. I suggest using a more systematic way for denoting all various species discussed in the paper, like it is used "TS" for the transition state. One of unexplained species is Pe₂, the reason for using Pe₄ for a key radical in the crosslinking mechanism is totally unclear. All radicals should be denoted by a dot, as it is in table 1 as s frequently used in the literature.
2. I suggest denoting respective reactions in Table 1 by corresponding numbers and use these numbering in Figure 2. At the same time, it is not clear which reactants participate in individual reactions in Figure 2 (some of them is just intramolecular isomerization). For example, in one of channels (red) is reactant denoted just as "H".
3. Last column in Table 1, means "dissociation energies of breaking bond in reactants" – in some reactions it is not clear which bond the authors mean (in some reactions it is just a rearrangement, in some there is the hydrogen abstraction ...). Apparently ΔH_0 and D_0 correspond to different processes? D_0 in reactions 3 to 5 are the same, 1.60 eV. This seems to correspond to the hydrogen abstraction reactions from the protonated benzophenone?
4. Some reactions in Figure 2 are characterized by quite high activation barriers. How such reactions, kinetically less favourable, can affect the cross-linking effectivity?
5. Headings of Tables and Figures should be self-explanatory. For example, Eg in Table 3 means the HOMO - LUMO energy gap; IP(a) and EA(a) are adiabatic (vertical v) values (?). Also, I suggest that HOMO - LUMO energy gaps and excitation energies should be presented in the same table. Negative sign for EA of Pe (first row in table 3) means that the corresponding anion is unstable?
6. One of key results reported in the paper is the finding that the PTAIC1 radical (in the notation used by the authors) can be crosslinked with the Pe (the polyethylene) radical effectively by the multiplication effect of cross-linking. The product will have quite different properties than crosslinked Pe-Pe. Which would be physical, chemical and structural aspects of different cross-linked polymers discussed in the present paper? Well organized parallel structures of cross-linked polyethylene can be possibly obtained by using metal atoms functioning as a "cross-linker".
7. English should be substantially improved at so many places that it would be wasting time mentioning all of them.

Review form: Reviewer 2

Is the manuscript scientifically sound in its present form?

Yes

Are the interpretations and conclusions justified by the results?

Yes

Is the language acceptable?

No

Is it clear how to access all supporting data?

No

Do you have any ethical concerns with this paper?

No

Have you any concerns about statistical analyses in this paper?

No

Recommendation?

Accept with minor revision (please list in comments)

Comments to the Author(s)

The reaction and activation energies must be the free Gibbs energies, not enthalpies. For the reactions where the change of the number of particles takes place the effect of entropy is important. This may notably change the energetics and the conclusions too. Since authors calculated ZPE corrections there will be no problems to present energy profiles in the form of the free Gibbs energy changes.

Also English should be revised. Sometimes the phrases are too long and difficult to understand.

Decision letter (RSOS-182196.R0)

24-Apr-2019

Dear Professor Zhang:

Title: Further Discussion on the Reaction Behavior of Triallyl Isocyanurate in the UV Radiation Cross-linking Process of Polyethylene: A Theoretical Study
Manuscript ID: RSOS-182196

The editor assigned to your manuscript has now received comments from reviewers. We would like you to revise your paper in accordance with the referee and Subject Editor suggestions which can be found below (not including confidential reports to the Editor). Please note this decision does not guarantee eventual acceptance.

Please submit your revised paper before 17-May-2019. Please note that the revision deadline will

expire at 00.00am on this date. If we do not hear from you within this time then it will be assumed that the paper has been withdrawn. In exceptional circumstances, extensions may be possible if agreed with the Editorial Office in advance. We do not allow multiple rounds of revision so we urge you to make every effort to fully address all of the comments at this stage. If deemed necessary by the Editors, your manuscript will be sent back to one or more of the original reviewers for assessment. If the original reviewers are not available we may invite new reviewers.

On behalf of the Subject Editor Professor Anthony Stace and the Associate Editor Professor Kim Jelfs.

RSC Associate Editor:
Comments to the Author:
(There are no comments.)

RSC Subject Editor:
Comments to the Author:
(There are no comments.)

Reviewers' Comments to Author:
Reviewer: 1

Comments to the Author(s)
The authors present impressive amount of data leading eventually to better understanding of more effective polyethylene (PE) crosslinking mechanism. The basic species is 4-methylheptane (Pe in the author's notation) serving as a model of cross-linkable polyethylene. The data include

the reaction energies, activation barriers, ionization and excitation energies and electron affinities of species supporting the effectivity of the crosslinking of polyethylene. From this point of view the paper is useful and interesting. Selected methods are adequate – for the geometry optimization, stationary point and transition states energies the common DFT/B3LYP method used in the present study is partly supported by previous successful applications to similar problems. Excitation energies are calculated by the TD DFT method. From this point of view no problem. The problem is that the presentation is rather poor. The paper is difficult to follow. Conclusions are not clearly supported by results. I would support publishing the paper only upon substantial improvement of the presentation.

1. I am lost in a jungle of abbreviations already while reading Part I, Introduction. Some are explained in Figure 1, mentioned much later (page 5), some remain unexplained at all. I suggest using a more systematic way for denoting all various species discussed in the paper, like it is used “TS” for the transition state. One of unexplained species is Pe₂, the reason for using Pe₄ for a key radical in the crosslinking mechanism is totally unclear. All radicals should be denoted by a dot, as it is in table 1 as s frequently used in the literature.
2. I suggest denoting respective reactions in Table 1 by corresponding numbers and use these numbering in Figure 2. At the same time, it is not clear which reactants participate in individual reactions in Figure 2 (some of them is just intramolecular isomerization). For example, in one of channels (red) is reactant denoted just as “H”.
3. Last column in Table 1, means “dissociation energies of breaking bond in reactants” – in some reactions it is not clear which bond the authors mean (in some reactions it is just a rearrangement, in some there is the hydrogen abstraction ...). Apparently ΔH_0 and D_0 correspond to different processes? D_0 in reactions 3 to 5 are the same, 1.60 eV. This seems to correspond to the hydrogen abstraction reactions from the protonated benzophenone?
4. Some reactions in Figure 2 are characterized by quite high activation barriers. How such reactions, kinetically less favourable, can affect the cross-linking effectivity?
5. Headings of Tables and Figures should be self-explanatory. For example, Eg in Table 3 means the HOMO – LUMO energy gap; IP(a) and EA(a) are adiabatic (vertical v) values (?). Also, I suggest that HOMO – LUMO energy gaps and excitation energies should be presented in the same table. Negative sign for EA of Pe (first row in table 3) means that the corresponding anion is unstable?
6. One of key results reported in the paper is the finding that the PTAIC1 radical (in the notation used by the authors) can be crosslinked with the Pe (the polyethylene) radical effectively by the multiplication effect of cross-linking. The product will have quite different properties than crosslinked Pe-Pe. Which would be physical, chemical and structural aspects of different cross-linked polymers discussed in the present paper? Well organized parallel structures of cross-linked polyethylene can be possibly obtained by using metal atoms functioning as a “cross-linker”.
7. English should be substantially improved at so many places that it would be wasting time mentioning all of them.

Reviewer: 2

Comments to the Author(s)

The reaction and activation energies must be the free Gibbs energies, not enthalpies. For the reactions where the change of the number of particles takes place the effect of entropy is important. This may notably change the energetics and the conclusions too. Since authors calculated ZPE corrections there will be no problems to present energy profiles in the form of the free Gibbs energy changes.

Also English should be revised. Sometimes the phrases are too long and difficult to understand.

Author's Response to Decision Letter for (RSOS-182196.R0)

See Appendix A.

RSOS-182196.R1 (Revision)

Review form: Reviewer 2

Is the manuscript scientifically sound in its present form?

No

Are the interpretations and conclusions justified by the results?

No

Is the language acceptable?

Yes

Do you have any ethical concerns with this paper?

No

Have you any concerns about statistical analyses in this paper?

No

Recommendation?

Accept with minor revision (please list in comments)

Comments to the Author(s)

The authors have not attended my comment to show the energy profile in a form of the Gibbs energy change, not the enthalpy. There is no need for the additional calculations for that, since authors have already done the frequency calculations. I insist that this is important. It follows from the formula $dG=dH-TdS$ that the enthalpy is a valid indicator for the reaction thermodynamics only if there is no entropy change during the reaction, and this is not the case. I must insist to present the reaction energy profiles in the form of the Gibbs energy change. The enthalpy changes are meaningless in this particular case. There is a possibility that the conclusions made by the authors based on the current reaction energy profiles when enthalpy is used are incorrect.

English is acceptable now.

Decision letter (RSOS-182196.R1)

12-Jun-2019

Dear Professor Zhang:

Title: Further Discussion on the Reaction Behavior of Triallyl Isocyanurate in the UV Radiation Cross-linking Process of Polyethylene: A Theoretical Study

Manuscript ID: RSOS-182196.R1

Thank you for submitting the above manuscript to Royal Society Open Science. On behalf of the Editors and the Royal Society of Chemistry, I am pleased to inform you that your manuscript will be accepted for publication in Royal Society Open Science subject to minor revision in accordance with the referee suggestions. Please find the reviewers' comments at the end of this email.

The reviewers and handling editors have recommended publication, but also suggest some minor revisions to your manuscript. Therefore, I invite you to respond to the comments and revise your manuscript.

Because the schedule for publication is very tight, it is a condition of publication that you submit the revised version of your manuscript before 21-Jun-2019. Please note that the revision deadline will expire at 00.00am on this date. If you do not think you will be able to meet this date please let me know immediately.

Once again, thank you for submitting your manuscript to Royal Society Open Science. The

chemistry content of Royal Society Open Science is published in collaboration with the Royal Society of Chemistry. I look forward to receiving your revision. If you have any questions at all, please do not hesitate to get in touch.

Best wishes,
Dr Laura Smith
Publishing Editor, Journals

On behalf of the Subject Editor Professor Anthony Stace and the Associate Editor Professor Kim Jelfs.

RSC Associate Editor:
Comments to the Author:
(There are no comments.)

RSC Subject Editor:
Comments to the Author:
(There are no comments.)

Reviewer comments to Author:
Reviewer: 2

Comments to the Author(s)

The authors have not attended my comment to show the energy profile in a form of the Gibbs energy change, not the enthalpy. There is no need for the additional calculations for that, since authors have already done the frequency calculations. I insist that this is important. It follows from the formula $dG=dH-TdS$ that the enthalpy is a valid indicator for the reaction thermodynamics only if there is no entropy change during the reaction, and this is not the case. I must insist to present the reaction energy profiles in the form of the Gibbs energy change. The enthalpy changes are meaningless in this particular case. There is a possibility that the conclusions made by the authors based on the current reaction energy profiles when enthalpy is used are incorrect.

English is acceptable now.

Author's Response to Decision Letter for (RSOS-182196.R1)

See Appendix B.

RSOS-182196.R2 (Revision)

Review form: Reviewer 1

Is the manuscript scientifically sound in its present form?

Yes

Are the interpretations and conclusions justified by the results?

Yes

Is the language acceptable?

Yes

Do you have any ethical concerns with this paper?

No

Have you any concerns about statistical analyses in this paper?

No

Recommendation?

Accept as is

Comments to the Author(s)

The revised manuscript is now acceptable

Decision letter (RSOS-182196.R2)

12-Aug-2019

Dear Professor Zhang:

Title: Further Discussion on the Reaction Behavior of Triallyl Isocyanurate in the UV Radiation Cross-linking Process of Polyethylene: A Theoretical Study
Manuscript ID: RSOS-182196.R2

It is a pleasure to accept your manuscript in its current form for publication in Royal Society Open Science. The chemistry content of Royal Society Open Science is published in collaboration with the Royal Society of Chemistry.

On behalf of the Subject Editor Professor Anthony Stace and the Associate Editor Professor Kim Jelfs.

RSC Associate Editor:
Comments to the Author:
(There are no comments.)

RSC Subject Editor:
Comments to the Author:
(There are no comments.)

Reviewer(s)' Comments to Author:
Reviewer: 1

Comments to the Author(s)
The revised manuscript is now acceptable

Appendix A

Dear Editor:

We are very grateful for your concern about our manuscript.

Title: Further Discussion on the Reaction Behavior of Triallyl Isocyanurate in the UV Radiation Cross-linking Process of Polyethylene: A Theoretical Study

Manuscript ID: RSOS-182196

The revision of this manuscript has been made according to the reviewers' comments, the changes have been highlighted by red in the revised version, and the point by point response is listed as follow:

=====
We would like to show our heartfelt thanks to the Reviewers for their helpful comments.

Reviewer: 1

The authors present impressive amount of data leading eventually to better understanding of more effective polyethylene (PE) crosslinking mechanism. The basic species is 4-methylheptane (Pe in the author's notation) serving as a model of cross-linkable polyethylene. The data include the reaction energies, activation barriers, ionization and excitation energies and electron affinities of species supporting the effectivity of the crosslinking of polyethylene. From this point of view the paper is useful and interesting. Selected methods are adequate – for the geometry optimization, stationary point and transition states energies the common DFT/B3LYP method used in the present study is partly supported by previous successful applications to similar problems. Excitation energies are calculated by the TD DFT method. From this point of view no problem. The problem is that the presentation is rather poor. The paper is difficult to follow. Conclusions are not clearly supported by results. I would support publishing the paper only upon substantial improvement of the presentation.

Comment 1

I am lost in a jungle of abbreviations already while reading Part I., Introduction. Some are explained in Figure 1, mentioned much later (page 5), some remain unexplained at all. I suggest using a more systematic way for denoting all various species discussed in the paper, like it is used "TS" for the transition state. One of unexplained species is Pe2, the reason for using Pe4 for a key radical in the crosslinking mechanism is totally unclear. All radicals should be denoted by a dot, as it is in table 1 as is frequently used in the literature.

Answer: Thanks a lot for the reviewer's kindly comment. The molecular formula and

corresponding abbreviations of the reactants and products of the studied molecules are listed in Table 1 in the revised version. The transition states and the products are abbreviated to the corresponding TS and P, respectively. The mentioned radical Pe2 have also been denoted in Table 1 in the revised version. About the issue, conclusions are not clearly supported by results, there are some explanations. In general, the results of theoretical chemistry calculation can explain microcosmic mechanism through analyzing electronic structure information and potential energy surface, infer the change of chemical bonds, and test the experimental hypothesis. In this paper, we try to investigate the possible reactions between Pe with additives (TAIC or TMPTMA) including side reactions during the polyethylene UV radiation cross-linking process. It play a crucial role for optimizing polyethylene UV radiation cross-linking process and developing insulation materials for high voltage cables. According the reviewer's comment, conclusions have been revised. The changes have been highlighted by red in the revised version.

Comment 2

I suggest denoting respective reactions in Table 1 by corresponding numbers and use these numbering in Figure 2. At the same time, it is not clear which reactants participate in individual reactions in Figure 2 (some of them is just intramolecular isomerization). For example, in one of channels (red) is reactant denoted just as "H".

Answer: Thanks a lot for the reviewer's kindly comment. In Figure 2, the other reactants have been introduced with plus sign "+" in the revised version. The reaction channels without plus sign are dissociations or intra-molecular isomerizations. We denote respective reactions in Table 2 in the revised version by corresponding numbers and use these numbering in Electronic Supplementary Materials (ESM-2).

Comment 3

Last column in Table 1, means "dissociation energies of breaking bond in reactants" – in some reactions it is not clear which bond the authors mean (in some reactions it is just a rearrangement, in some there is the hydrogen abstraction ...). Apparently ΔH_0 and D_0 correspond to different processes? D_0 in reactions 3 to 5 are the same, 1.60 eV. This seems to correspond to the hydrogen abstraction reactions from the protonated benzophenone?

Answer: The reaction enthalpies (ΔH_{298}^0) and the potential barrier heights (ΔE^{TS}) with zero-point energy (ZPE) corrections are correspond to different reaction channels. The

dissociation energies of breaking bond in reactants are listed in the last column. The dissociation energies of breaking bond in reactants of H addition reactions haven't been listed, denoting as "--". The dissociation energies in reactions 3 to 5 are the same, 1.60 eV, it means the dissociation energy of breaking bond O—H in reactant PBp is 1.60 eV, no matter which reaction it is, hydrogen dissociation from PBp (reaction 3), hydrogen abstraction from PBp by TAIC (reaction 4), or hydrogen abstraction from PBp by TMPTMA (reaction 5).

Comment 4

Some reactions in Figure 2 are characterized by quite high activation barriers. How such reactions, kinetically less favourable, can affect the cross-linking effectivity?

Answer: The main materials for preparing cross-linked polyethylene (XLPE) generally include low density polyethylene (LDPE), benzophenone (Bp, as photoinitiator), antioxidants, and triallyl isocyanurate (TAIC, as multi-functional crosslinker), etc. When the main materials pass through the UV irradiation system composed of the high power UV lamps, a lot of complicated reactions may occur. Accompanied by a lot of side reactions, many by-products appeared. These by-products affect performance of XLPE. To elucidate chemical reactions possibly during cross-linking of polyethylene play a crucial role in optimizing polyethylene UV radiation cross-linking process and developing insulation materials for high voltage cables. In this paper, we try to investigate on the possibility of these reactions and the dominant reaction channel during the polyethylene UV radiation cross-linking process. New experimental and theoretical efforts are required.

Comment 5

Headings of Tables and Figures should be self-explanatory. For example, Eg in Table 3 means the HOMO – LUMO energy gap; IP(*a*) and EA(*a*) are adiabatic (vertical *v*) values (?).

Also, I suggest that HOMO – LUMO energy gaps and excitation energies should be presented in the same table. Negative sign for EA of Pe (first row in table 3) means that the corresponding anion is unstable?

Answer: Thanks a lot for the reviewer's kindly comment. Heading of Table 3 has been complemented. *v* and *a* represent vertical energy and adiabatic energy, respectively. HOMO–LUMO energy gaps and excitation energies have been presented in Table 3 in the revised version. Negative sign of calculation obtained for EA of Pe means that the

Pe molecule can't capture electron.

Comment 6

One of key results reported in the paper is the finding that the PTAIC1 radical (in the notation used by the authors) can be crosslinked with the Pe (the polyethylene) radical effectively by the multiplication effect of cross-linking. The product will have quite different properties than crosslinked Pe-Pe. Which would be physical, chemical and structural aspects of different cross-linked polymers discussed in the present paper? Well organized parallel structures of cross-linked polyethylene can be possibly obtained by using metal atoms functioning as a “cross-linker”.

Answer: The PTAIC1 radical can be crosslinked with the Pe (4-methylheptane, as a model of cross-linkable polyethylene) radical effectively by the cross-linking. One TAIC molecule can connect three polyethylene macro molecules, and the TAIC molecule was introduced in cross linking network. This kind of small molecules involvement cause many differences in material properties chemically and physically. The introduction of TAIC molecules in cross-linking network introduced some polar group in the material such as carbonyl group and nitrogen. The material became more or less polar from nonpolar. This will change some properties of the material. For instance, the uniformly distributed and position fixed polar group during cross-linking process may form, it would act as traps to capture the electrical charge carriers, and form Coulomb force field to reject further charges transportation and suppress the electrical conductivity of the material eventually. The following reports will present the utilization of these properties. Further work to obtain well organized parallel structures of cross-linked polyethylene by using metal atoms functioning as a “cross-linker” will be considered later.

Comment 7

English should be substantially improved at so many places that it would be wasting time mentioning all of them.

Answer: Thanks a lot for the reviewer's careful comment. We checked carefully the writing and the grammar throughout the paper. The corresponding mistakes have been revised. The changes have been highlighted by red in the revised version.

Reviewer: 2

Comment

The reaction and activation energies must be the free Gibbs energies, not enthalpies. For the reactions where the change of the number of particles takes place the effect of entropy is important. This may notably change the energetics and the conclusions too. Since authors calculated ZPE corrections there will be no problems to present energy profiles in the form of the free Gibbs energy changes.

Also English should be revised. Sometimes the phrases are too long and difficult to understand.

Answer: Thanks a lot for the referee's comment. The reaction and activation energies would be the free Gibbs energies in further researches. We checked carefully the writing and the grammar throughout the paper. Those long phrases have been simplified in the revised version. The corresponding mistakes have been revised. The changes have been highlighted by red in the revised version.

Appendix B

Dear Editor:

We are very grateful for your concern about our manuscript.

Title: Further Discussion on the Reaction Behavior of Triallyl Isocyanurate in the UV Radiation Cross-linking Process of Polyethylene: A Theoretical Study

Manuscript ID: RSOS-182196.R1

Comment

The authors have not attended my comment to show the energy profile in a form of the Gibbs energy change, not the enthalpy. There is no need for the additional calculations for that, since authors have already done the frequency calculations. I insist that this is important. It follows from the formula $dG=dH-TdS$ that the enthalpy is a valid indicator for the reaction thermodynamics only if there is no entropy change during the reaction, and this is not the case. I must insist to present the reaction energy profiles in the form of the Gibbs energy change. The enthalpy changes are meaningless in this particular case. There is a possibility that the conclusions made by the authors based on the current reaction energy profiles when enthalpy is used are incorrect.

English is acceptable now.

Answer: The revision of this manuscript has been made according to the reviewer's comment. The changes have been highlighted in red color in the revised version. The reaction Gibbs free energies at 298 K (ΔG) and the Gibbs potential barrier heights (ΔG^\ddagger) including zero-point energy (ZPE) corrections of the thirteen reaction channels have been listed in Table 2 and the relevant schematic potential energy surfaces for the thirteen reaction channels have been adjusted in Figure 2 in the revised version.